# Data Normalization of Urine miRNA Profiling from Head and Neck Cancer Patients Treated with Cisplatin

**DOI:** 10.3390/ijms241310884

**Published:** 2023-06-29

**Authors:** Nadine de Godoy Torso, Julia Coelho França Quintanilha, Maria Aparecida Cursino, Eder de Carvalho Pincinato, Carmen Silvia Passos Lima, Patricia Moriel

**Affiliations:** 1School of Medical Sciences, University of Campinas, Campinas 13083894, Brazil; nadinetorso@gmail.com (N.d.G.T.); jcfquintanilha@gmail.com (J.C.F.Q.); ma.cursino@gmail.com (M.A.C.); eder.pincinato@unicamp.br (E.d.C.P.); carmenl@unicamp.br (C.S.P.L.); 2Faculty of Pharmaceutical Sciences, University of Campinas, Campinas 13083970, Brazil

**Keywords:** microRNA, qRT-PCR, endogenous normalizer, reference gene, cisplatin

## Abstract

The microRNA (miRNA) expression profile by qRT-PCR depends directly on the most appropriate normalization strategy adopted; however, currently there is no universally adequate reference gene. Therefore, this study aimed to determine, considering RNA-Seq results, the most adequate endogenous normalizer for use in the relative quantification of urine miRNAs from head and neck cancer patients, treated with cisplatin chemoradiotherapy. The massive sequencing was performed to identify the miRNAs differentially expressed between the group with cisplatin nephrotoxicity (*n* = 6) and the one without (*n* = 6). The candidate endogen normalizer was chosen according to four criteria: (1) the miRNA must be expressed in most samples; (2) the miRNA must have a fold change value between 0.99 and 1.01; (3) the miRNA must have a *p*-value ≥ 0.98; and (4) the miRNA must not be commented on by the final GeneGlobe (Qiagen, Hilden, Germany) analysis. Four miRNAs met all the criteria (hsa-miR-363-5p, hsa-miR-875-5p, hsa-miR-4302, and hsa-miR-6749-5p) and were selected for validation by qRT-PCR in a cohort of 49 patients (including the 12 sequencing participants). Only hsa-miR-875-5p was shown to be an adequate normalizer for the experimental condition under investigation, as it exhibited invariant expression between the two groups.

## 1. Introduction

There is a growing number of studies evaluating the expression profile of microRNAs (miRNAs) in different pathological processes [1], because one of their essential functions is to regulate gene expression at the molecular level. Currently, the miRNA expression profile by quantitative Real-Time PCR (qRT-PCR) is considered the gold standard for small RNAs, because it has high sensitivity and specificity [2]. This approach usually involves data normalization, where the expression level of each target gene is normalized to the expression of a stable internal reference gene known as a normalizer. It is a mathematical process that aims to remove those variations that are not consequences of the biological condition studied (including variations in extraction, reverse transcription, or amplification yield), to identify appropriate expression differences between the groups of samples [3,4].

However, as the integrity of the results obtained through qRT-PCR depends directly on the normalization strategy [5,6], it represents an important challenge for analyzing miRNAs in biofluids. To date, there is no definitive guidance or recommended normalization; there is also no universally adequate reference gene. So, when adopting different normalization strategies, different studies may find misinterpretations of biological effects.

There are several reasons for the absence of such a consensus, but the most significant is that miRNA expression patterns vary under different experimental or physiological conditions [7]. Numerous studies have shown that the potential reference gene expression levels differ considerably across different tissues and pathological conditions, making them inappropriate for normalization purposes [8,9,10].

One of the most adopted normalization strategies is the use of constitutive miRNAs that have been previously reported to be stably expressed across tissue and cell types [11], serving as an internal control in quantification. Some examples already used in previous works include miR-16, miR-103, miR-191, and RNU6B [4,12,13]; however, it is not possible to assure that these miRNAs will always have a constant pattern of expression under different biological or experimental conditions [11,14,15]. Further, as the extraction efficiency of miRNAs is smaller than the efficiency of longer RNAs, it is important to choose a normalizer with comparable properties as the quantified target [4].

Another commonly used miRNA is cel-miR-39, a miRNA from *Caenorhabditis elegans* commonly spiked into the samples because it does not have sequence homology to human miRNAs [16]. Nevertheless, as it is a miRNA of exogenous origin, this is not the most suitable option for normalization [7]. In fact, it is a good option to be used only as a quality control of miRNA extraction efficiency (as the amount of cel-miR-39 added to the samples is known and the amount recovered after extraction is directly related to the extraction efficiency [11,17]).

So, to avoid incorrect conclusions, it is essential to know the most appropriate normalizer to be used in the experimental condition under investigation. Despite significant advances in miRNA research, endogenous normalizers have not yet been described for use in urine miRNA expression. Therefore, the present study discusses this issue and aims to determine, taking into account RNA-Seq analysis, the most adequate endogenous normalizer for use in the relative quantification of urine miRNAs from head and neck cancer patients treated with cisplatin chemoradiotherapy.

## 2. Results

### 2.1. Participants

The criterion adopted for the subdivision of patients was whether or not they had nephrotoxicity, characterized as an increase in serum creatinine (SCr) ≥ 2 five days after (D5) the first chemotherapy cycle, according to the Common Toxicity Criteria for Adverse Events (CTCAE) version 4 [18]. So, massive sequencing was performed to identify the miRNAs differentially expressed between two groups: (1) the group with cisplatin nephrotoxicity (*n* = 6, participants who presented an increase in SCr ≥ grade 2 on D5, by CTCAE); (2) the group without cisplatin nephrotoxicity (*n* = 6, participants who presented an increase in SCr = 0 on D5, according to CTCAE) (Figure 1). Grade ≥ 2 increased SCr is defined as an increase of more than two times in SCr baseline values.

The clinical characteristics of all participants included in this study have been previously published [19] and are reproduced in Table 1; Table 2 shows their nephrotoxicity parameters. The mean age in the two groups was quite similar, as was the proportion between males and females. The comparison of renal function between the two groups five days after the first cycle of chemotherapy showed statistically significant differences both for laboratory parameters (change in serum creatinine D5—baseline, *p* = 0.005; change in D5 creatinine clearance—baseline, *p* = 0.005) and for the adopted acute kidney injury (AKI) classifications (Risk, Injury, Failure, Loss, and End-stage kidney disease—RIFLE, *p* = <0.005; and Acute Kidney Injury Network, AKIN, *p* = <0.005).

### 2.2. MiRNA Sequencing

For each miRNA found in the sequencing, the GeneGlobe Data Analysis Center (Qiagen) provides the fold change (fold change = miRNA expression of the nephrotoxicity group/miRNA expression of the non-nephrotoxicity group), fold regulation (if fold change ≥ 1, fold regulation = fold change; if fold change < 1, fold regulation = 1/fold change), and *p*-value calculated based on the Wald test. Table 3 shows the sequencing results for the urinary miRNAs that presented a fold change between 0.99 and 1.01.

Of the 20 miRNAs shown in this table, only four met all the criteria established in this study for defining possible normalizer candidates, namely, hsa-miR-363-5p, hsa-miR-875-5p, hsa-miR-4302, and hsa-miR-6749-5p.

Figure 2 presents the Volcano plots with all urinary miRNAs identified in the sequencing results, highlighting those four selected as possible candidates for normalizers. 

### 2.3. Validation of Candidates for Normalizing miRNAs

The four possible options highlighted in Table 3 (hsa-miR-363-5p, hsa-miR-875-5p, hsa-miR-4302, and hsa-miR-6749-5p) were selected for validation by qRT-PCR in a cohort of 49 patients (*n* = 49, including the 12 sequencing participants). 

Despite having been identified by the sequencing technique in all 12 samples used, the amplification of hsa-miR-363-5p and hsa-miR-6749-5p by the qRT-PCR technique was not verified. Several tests were performed using cDNA samples from the same sequencing participants and varying the amount of cDNA used, but amplification was still not verified. 

As for the other two endogenous miRNAs, Table 4 and Table 5 show the expression of hsa-miR-875-5p and hsa-miR-4302, respectively, both at baseline and at D5. Regarding hsa-miR-875-5p, it is possible to verify that its expression remained similar in both groups both in the baseline period (*p* = 0.5354, Wald test) and on D5 (*p* = 0.4418, Wald test) (*n* = 49), thus fulfilling the basic characteristics to be considered an adequate endogenous normalizer.

However, about hsa-miR-4302, its expression varied significantly between the groups with and without cisplatin nephrotoxicity at the baseline period (*p* < 0.05, Wald test). For this reason, this miRNA cannot be considered a good endogenous normalizer.

Figure 3 presents the comparative expression of hsa-miR-875-5p and has-miR-4302 in the cohort of 49 participants. 

### 2.4. Stability Assessment of Selected Endogenous miRNAs

The algorithms comprised in the RefFinder tool work by first estimating the stability value of a specific reference gene (using data from four different tools) and then comparing it to the stability values of other endogenous controls: the lower the stability value, the more appropriate the endogenous control is [23].

In the present study, the general analysis performed by RefFinder indicates miR-4302 as the most stable candidate (Figure 4), although its expression differed significantly between the two patient groups examined.

## 3. Discussion

For a reference gene to be considered universally adequate, its expression level would need to remain constant in all experimental conditions and experimental groups [24]; however, this uniform condition does not exist. Hence, it is recommended that the efficiency of reference genes as a normalization strategy be verified for the specific tissue or sample type, as well as for the experimental conditions; so, this study aimed to verify the best normalizer for urinary miRNA expression in the context of head and neck cancer patients treated with cisplatin chemoradiotherapy. 

Numerous strategies have been previously proposed to normalize data from qRT-PCR for miRNAs of plasma origin; however, this number is even scarcer for studies of urinary miRNAs. Whether for one sample or another, to date, there has been no consensus on the reference miRNA that should be used. For example, many studies still use creatinine as a normalizer when evaluating the expression profile of urinary miRNAs [25]. However, especially in the case of AKI patients (the participants considered in this study), it is known that creatinine has important deficiencies in terms of its specificity and sensitivity [26]. For this reason, finding a normalizer in the AKI context is extremely relevant.

Therefore, the present study chose to use the RNA-sequencing technique, a high-throughput assay which provides a complete overview for miRNA expression profiling. Although commonly used for screening purposes (where the objective usually is to identify miRNAs that are differentially expressed between biofluids, tissues, or subgroups of patients [27]), this technique can also be used to select the most adequate candidate normalizer miRNAs (as was the case in this study, where the objective was to identify the miRNAs whose expression pattern was equivalent across all samples in both groups of patients evaluated, those who did and those who did not develop cisplatin-induced nephrotoxicity). 

First, the two groups evaluated were shown to have significant differences for AKI biomarkers and classifications considered (increase in SCr, creatinine clearance, RIFLE, and AKIN; *p* < 0.005) but not for clinical variables (including age, gender, and Karnofsky Performance Status; *p* > 0.005). This was important to ensure that the profile difference between the two groups did not interfere with the evaluation of the expression of possible normalizing miRNAs.

Using the four criteria, data from the RNA-Seq (containing 20 miRNAs with fold change between 0.99 and 1.01) resulted in the selection of four miRNAs for evaluation by qRT-PCR: hsa-miR-363-5p, hsa-miR-6749-5p, hsa-miR-875-5p, and hsa-miR-4302. Nonetheless, the amplification by qRT-PCR of the first two mentioned was not verified; it may have happened because qRT-PCR and sequencing techniques are not always considered comparable (which can also be said about their results) [28]. Stokowy et al. have shown that qRT-PCR may have limitations in validating high-throughput sequencing data, including a lower specificity for detecting one-base changes in the target sequence [28]. As qRT-PCR is considered the gold standard in transcript quantification, in this study, hsa-miR-363-5p and hsa-miR-6749-5p were not considered suitable options for candidate endogenous normalizers.

Further, the expression of hsa-miR-4302 varied significantly between the two studied groups at the baseline period (*p* < 0.05, Wald test). For this reason, this miRNA also cannot be considered for use as an endogenous normalizer.

The last of these four miRNAs, hsa-miR-875-5p, showed that its expression remained similar in both groups both in the baseline period (*p* = 0.5354, Wald test) and on D5 (*p* = 0.4418, Wald test) (*n* = 49). Thus, the use of this miRNA as an endogenous normalizer of urinary miRNA profiling seems to ensure that its expression is stable between the two groups of participants. The few works already published involving hsa-miR-875-5p have shown only its involvement in tumorigenesis of different histological origins [29,30,31]; so, for its role as an endogenous normalizer to be confirmed for urinary miRNA profiling, it would be interesting for future studies to assess whether it is really not involved in AKI. Nevertheless, for this miRNA to be used also as an endogenous normalizer in other contexts, it must be verified that it is not altered by the treatment received or the disease under study. Otherwise, the future results obtained could not be trusted.

After identifying which genes are likely candidates, it is necessary to verify their stability. According to RefFinder analysis, the most stable candidate among the two considered is miR-4302, as it exhibits the smallest stability value. However, as its expression differed significantly throughout the two investigated sample sets, hsa-miR-875-5p has to be considered the most appropriate normalizing gene for urine miRNA expression profiling in head and neck cancer patients treated with cisplatin chemoradiotherapy.

### Future Perspectives

One of the recommendations in choosing the ideal normalizer is to use one selected from sequencing on the same sample source, because the content and concentration profile of miRNAs vary greatly according to the human biofluid [32]. Therefore, it is expected that novel studies will be conducted with other biofluids and under other disease conditions.

In addition, it would be equally interesting for future studies to investigate whether the most suitable method for normalization is the use of a specific miRNA or a set of them [14]. 

Finally, it is suggested that, henceforth, when further information about these miRNAs becomes available (including published work on their functional role, particularly whether they have any influence on the development of AKI), the expression of hsa-miR-4302 and hsa-miR-6749-5p as candidates for endogenous normalizers is also verified.

## 4. Materials and Methods

### 4.1. Ethical Considerations

This study was conducted in accordance with the Declaration of Helsinki and was approved by the Ethics Committee of the University of Campinas (protocol code: 65397517.7.0000.5404, 8 February 2021). All participants signed the informed consent form.

### 4.2. Participants and Treatment Regimen

This study was conducted in the Clinical Oncology Department of the *Hospital de Clínicas*, University of Campinas (HC-UNICAMP), a large tertiary teaching hospital in Campinas, Brazil.

The study population and data collection have been previously described [19]. Briefly, participants were included in the study according to the following criteria: aged between 18 and 80 years old; diagnosed with primary head and neck squamous cell carcinoma, evidenced by biopsy report; and who received, as a therapeutic approach, antineoplastic treatment based on cisplatin (doses ranged from 80–100 mg/m^2^; administered in three cycles, every 21 days) concurrently with radiotherapy (total dose of 70 Gy, divided into 35 daily applications of 2 Gy administered 5 days/week for 7 weeks). 

On the day of chemotherapy, as part of the protocol for the prevention of adverse reactions, the patients also received vigorous hydration, diuresis, and prophylaxis for acute emesis. For treatment and prevention of delayed emesis, patients received 10 mg of metoclopramide every 6 h and 8 mg of dexamethasone every 12 h for three consecutive days after the chemotherapy sessions.

### 4.3. Nephrotoxicity Assessment

Cisplatin-induced nephrotoxicity was assessed through blood collections performed before and on D5; baseline values were compared to values found after cisplatin administration. Laboratory markers analyzed for nephrotoxicity were SCr, estimated creatinine clearance (estimated using the Cockcroft-Gault formula [33]), urea, sodium, magnesium, calcium, phosphorus, and potassium; changes in these markers from their baseline values were graded for severity according to the CTCAE version 4 [18]. AKI was also classified according to the Risk, Injury, Failure, Loss, and End-stage kidney disease (RIFLE) [34] and Acute Kidney Injury Network (AKIN) [35] criteria.

### 4.4. MiRNA Sequencing

Urine samples collected on D5 were used; the choice of these collection times was based on the work of Pavkovic et al., which suggests maximum miRNA changes after five days of cisplatin administration [36]. After collection, the samples were centrifuged at 2500 rpm at 4 °C for 10 min; the supernatant was aliquoted and stored in a −80 °C freezer until the time of analysis. Details of miRNA sequencing methods have been previously reported [19].

The sequencing cohort consisted of 12 participants (*n* = 12), while the validation cohort comprised 49 participants (*n* = 49, including the 12 sequencing participants).

### 4.5. Quantitative Real-Time PCR for the Complete Cohort

For miRNA expression assessment in the complete cohort (*n* = 49), miRNAs were extracted from urine samples collected at baseline and on D5, using the miRNeasy Serum/Plasma Kit (Qiagen, Germany), following the manufacturer’s instructions. During extraction procedures, 3 × 10^7^ copies of the synthetic miRNA *Caenorhabditis elegans* miR-39 (cel-miR-39) were added to be used as an exogenous control (spike-in). cDNA was synthesized using the TaqMan™ Advanced miRNA cDNA Synthesis Kit (Applied Biosystems, Waltham, MA, USA), following the manufacturer’s instructions.

RT-qPCR reactions were performed on a Rotor-Gene Q (Qiagen, Germany), using TaqMan™ Advanced miRNA Assays (Applied Biosystems, Waltham, MA, USA) for the miRNAs selected as possible endogenous normalizers, as well as for cel-miR-39 (spike-in) [28]. The total reaction volume was reduced to 10 µL, consisting of 5 µL of TaqMan^®^ Fast Advanced Master Mix (2×) (Applied Biosystems, USA), 0.5 µL of TaqMan^®^ Advanced miRNA Assay (20×) (Applied Biosystems, USA), 2 µL of RNase-free water, and 2.5 µL of diluted cDNA (1:10). Raw data were evaluated using Rotor-Gene Q Series Software 2.1.0.9 (Qiagen, Germany).

As part of the quality control, samples whose cel-miR-39 expression was above two standard deviations were excluded from the analysis. 

### 4.6. Criteria for Defining Possible Normalizer Candidates

Considering the RNA-Seq expression data, four criteria were used to select the candidate endogenous normalizer miRNAs: (1) the miRNA must be expressed in most, if not all, of the samples; (2) the miRNA must have a fold change value between 0.99 and 1.01; (3) the miRNA must have a *p*-value ≥ 0.98; and (4) the miRNA must not be commented on by the final GeneGlobe analysis.

### 4.7. Statistical Analysis

For the analysis of clinical and demographic data, absolute frequencies/percentages and measures of position (mean) and dispersion (standard deviation) are presented. Nephrotoxicity markers and participant profiles in the two groups studied were compared at baseline and D5 using the Mann–Whitney test or Fisher’s exact test. Data normality was tested using the Shapiro–Wilk test. In the validation phase, miRNA expressions were compared between groups with and without cisplatin-induced nephrotoxicity at baseline and D5 using the Mann–Whitney test. A value of *p* < 0.05 was considered statistically significant for all analyses. The stability of endogenous control miRNAs was assessed using the RefFinder online tool (http://blooge.cn/RefFinder/); it comprises four different commonly used normalization tools, including BestKeeper, Comparative DeltaCt, NormFinder, and GeNorm, which apply different algorithms to determine the most consistently expressed gene. All statistical analyses were performed using GraphPad Prism v.9.1.0 software for Windows (GraphPad Software, Inc., San Diego, CA, USA).

## 5. Conclusions

The use of the most appropriate normalization strategy for each case is extremely important because failure in this choice can ultimately lead to wrong conclusions. In this sense, the main concern of this work was to identify a urinary miRNA whose expression was comparable in both participant groups (i.e., unaltered even with cisplatin-induced nephrotoxicity). Therefore, considering that hsa-miR-875-5p exhibits invariant expression between the two groups, it seems to be the most appropriate miRNA to be used as a normalizer in urine miRNA profiling of head and neck cancer patients treated with cisplatin chemoradiotherapy.

Finally, when compared to previously published studies evaluating candidates for normalizers for miRNAs of plasma origin, studies for urinary miRNAs are still scarce. Therefore, it is expected that this work can contribute to the growth of this scenario.

## Figures and Tables

**Figure 1 ijms-24-10884-f001:**
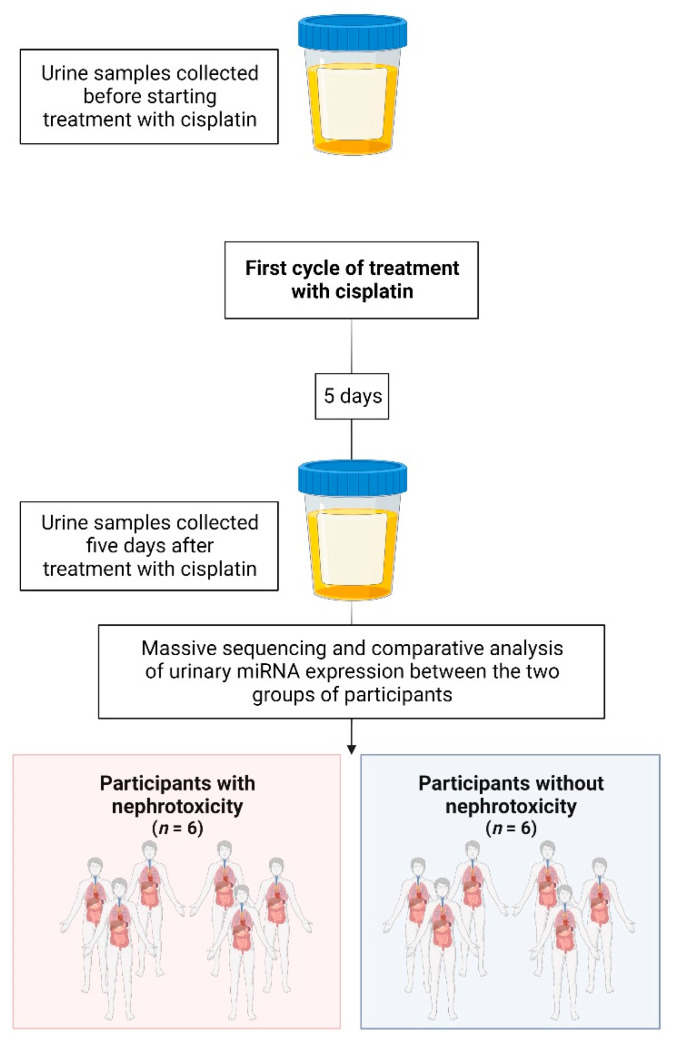
Samples collected and participant division: (1) group with cisplatin nephrotoxicity (*n* = 6, participants who presented an increase in SCr ≥ 2 on D5, by CTCAE); (2) group without cisplatin nephrotoxicity (*n* = 6, participants who presented an increase in SCr = 0 on D5, according to CTCAE) [18]. Grade ≥ 2 increased SCr is defined as an increase of more than two times in SCr baseline values. (Created with BioRender.com, accessed on 16 January 2023).

**Figure 2 ijms-24-10884-f002:**
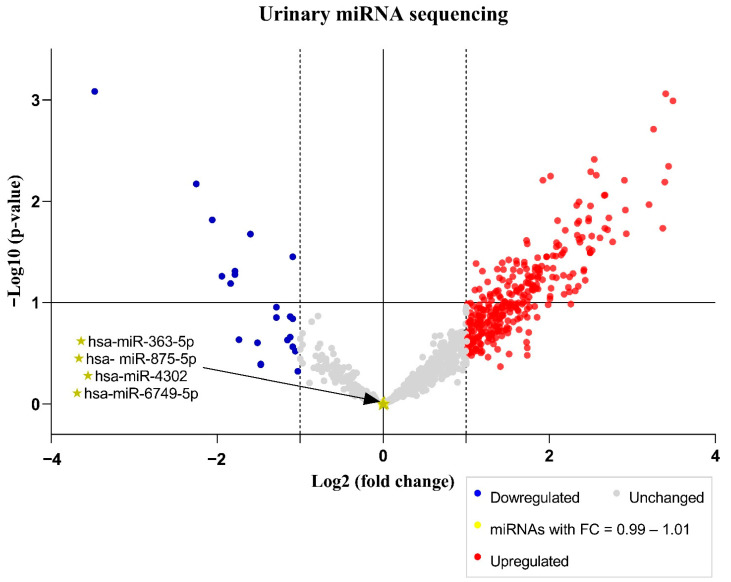
Volcano plot of expression of miRNAs in the urine of head and neck cancer patients treated with cisplatin chemoradiotherapy. The least expressed miRNAs in the cisplatin-induced nephrotoxicity group are shown as blue dots, and the most expressed miRNAs in the cisplatin-induced nephrotoxicity group are shown as red dots; the miRNAs that had no difference in expression between the two groups are shown as gray dots. miRNAs with fold change = 0.99–1.01 are shown as yellow dots (these are the miRNAs mentioned in Table 3); the four miRNAs selected as possible candidates for normalization based on the four criteria (expressed in most samples; fold change = 0.99–1.01; *p*-value ≥ 0.98; no comment by the final GeneGlobe analysis) are shown as a gold star. FC, fold change = miRNA expression of the nephrotoxicity group/miRNA expression of the non-nephrotoxicity group.

**Figure 3 ijms-24-10884-f003:**
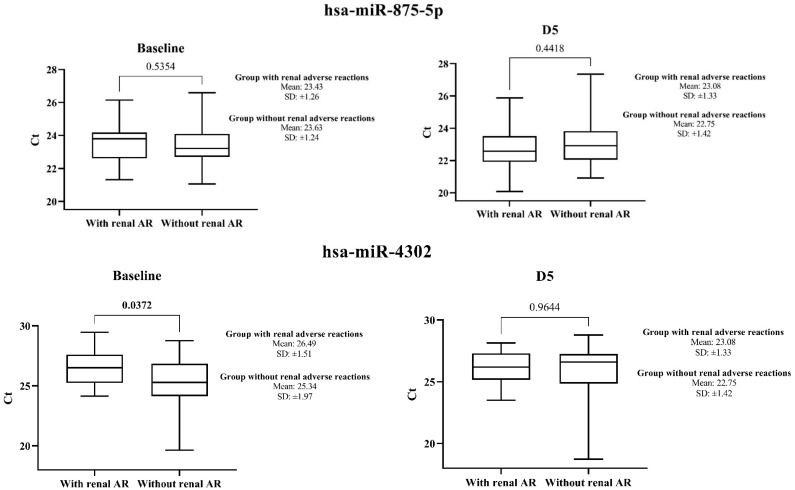
Box plot of urinary hsa-miR-875-5p and has-miR-4302 expression (in absolute Ct values), validated by qRT-PCR in participants with cisplatin-induced nephrotoxicity (*n* = 24) and participants who did not have cisplatin-induced nephrotoxicity (*n* = 25). AR, adverse reaction; Ct, threshold cycle; D5, 5th day after the first cycle of cisplatin chemotherapy; FC, fold change = miRNA expression of the nephrotoxicity group/miRNA expression of the non-nephrotoxicity group; SD, standard deviation.

**Figure 4 ijms-24-10884-f004:**
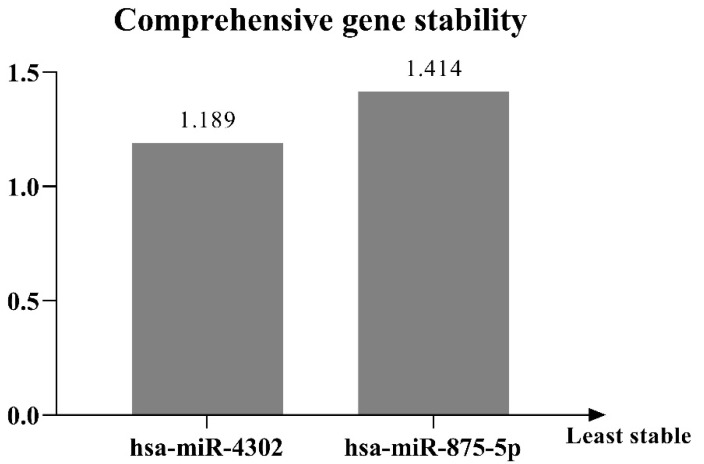
General stability analysis of candidate endogenous miRNAs by RefFinder (http://blooge.cn/RefFinder/).

**Table 1 ijms-24-10884-t001:** Participants’ clinical data whose urinary miRNA expression profiles were evaluated in the sequencing (*n* = 12).

Variable	Participants with Cisplatin-Induced Nephrotoxicity (*n* = 6)	Participants without Cisplatin-Induced Nephrotoxicity (*n* = 6)
**Age at diagnosis (mean ± SD, years)**	57.50 ± 7.40	60.33 ± 4.89
**Gender (*n*, %)**		
Male	4 (66.7)	5 (83.3)
Female	2 (33.3)	1 (16.7)
**Ethnicity (*n*, %)**		
Caucasian	6 (100)	4 (66.7)
Non-Caucasian	0	2 (33.3)
**Smoking category** [20,21] **(*n*, %)**		
Never smoked	1 (16.7)	1 (16.7)
Light smoker	0 (0)	0
Moderate smoker	1 (16.7)	0
Heavy smoker	4 (66.7)	5 (83.3)
**Drinking category** [22] **(*n*, %)**		
Abstainer	2 (33.3)	2 (33.3)
Light drinker	0	0
Moderate drinker	0	0
Heavy drinker	1 (16.7)	3 (50.0)
Very heavy drinker	3 (50.0)	1 (16.7)
**Tumor site (*n*, %)**		
Oral cavity	1 (16.7)	3 (50.0)
Larynx	1 (16.7)	1 (16.7)
Hypopharynx	3 (50.0)	2 (33.3)
Oropharynx	1 (16.7)	0
**Tumor stage (*n*, %)**		
I	0	0
II	0	0
III	0	1 (16.7)
IV	6 (100)	5 (83.3)
**KPS (*n*, %)**		
90	3 (50.0)	4 (66.7)
80	3 (50.0)	2 (33.3)

KPS, Karnofsky Performance Status; *n* absolute number of patients; SD, standard deviation.

**Table 2 ijms-24-10884-t002:** Results of renal biomarkers and AKIN and RIFLE classifications for the 12 participants.

	Participants with Cisplatin-Induced Nephrotoxicity (*n* = 6)		Participants without Cisplatin-Induced Nephrotoxicity (*n* = 6)		*p*-Value
	Baseline	D5	Baseline	D5	
**SCr (mg/dL)** **(mean ± SD)**	0.7 ± 0.1	3.4 ± 2.1	0.9 ± 0.2	0.9 ± 0.2	-
**SCr variation (mg/dL)** **(D5—Baseline; mean ± SD)**	2.7 ± 2.1		0.0 ± 0.1		**0.005 ^a^**
**Creatinine clearance (mL/min)** **(mean ± SD)**	97.6 ± 40.5	25.9 ± 11.5	70.2 ± 22.9	69.7 ± 24.0	-
**Creatinine clearance variation (mL/min)** **(D5—Baseline; mean ± SD)**	−71.8 ± 38.5		−0.5 ± 6.6		**0.005 ^a^**
**CTCAE—Increased SCr (D5)** **(*n*, %)**					
Grade 0	0		6 (100)		**0.002 ^b^**
Grade 1	0		0		
Grade 2	3 (50.0)		0		
Grade 3	3 (50.0)		0		
**CTCAE—Creatinine clearance (D5)** **(*n*, %)**					
Grade 0	0		2 (33.3)		**0.015 ^b^**
Grade 1	0		2 (33.3)		
Grade 2	3 (50.0)		1 (16.7)		
Grade 3	1 (16.7)		0		
Grade 4	2 (33.3)		0		
**RIFLE (D5)** **(*n*, %)**					
AKI not determined by this criterion	0		6 (100)		**0.002 ^b^**
Risk (R)	0		0		
Injury (I)	3 (50.0)		0		
Failure (F)	3 (50.0)		0		
**AKIN (D5)** **(*n*, %)**					
AKI not determined by this criterion	0		6 (100)		**0.002 ^b^**
Grade 1	3 (50.0)		0		
Grade 3	3 (50.0)		0		

AKI, acute kidney injury; AKIN, Acute Kidney Injury Network; CTCAE, Common Terminology Criteria for Adverse Events (assessment according to increase in serum creatinine); D5, 5th day after the first cycle of cisplatin chemotherapy; *n*, absolute number of patients; RIFLE, Risk, Injury, Failure, Loss, and End-stage kidney disease; Scr, serum creatinine; SD, standard deviation; ^a^ Mann–Whitney test; ^b^ Fisher’s exact test.

**Table 3 ijms-24-10884-t003:** Sequencing results for the urinary miRNAs with fold changes between 0.99 and 1.01 (miRNAs are ordered in the table in ascending numerical order according to their nomenclature). The four miRNAs identified as potential normalizers (according to the criteria considered in this study) are highlighted in bold.

miRNA	Fold Change	*p*-Value ^a^	Comments by the GeneGlobe Analysis
hsa-miR-136-3p	1.01	0.988	A
hsa-miR-196b-5p	1.00	0.995	A
**hsa-miR-363-5p**	**1.01**	**0.982**	-
hsa-miR-494-3p	1.01	0.987	A
hsa-miR-668-3p	0.99	0.992	B
hsa-miR-668-5p	1.01	0.996	B
hsa-miR-873-5p	0.99	0.995	B
**hsa-miR-875-5p**	**1.00**	**0.999**	**-**
hsa-miR-1237-5p	0.99	0.981	B
hsa-miR-3689e	1.00	1.000	B
hsa-miR-3692-5p	0.99	0.988	B
hsa-miR-3944-3p	1.01	0.981	A
**hsa-miR-4302**	**0.99**	**0.987**	**-**
hsa-miR-4700-5p	0.99	0.986	A
hsa-miR-6506-5p	1.00	0.997	B
**hsa-miR-6749-5p**	**1.00**	**0.999**	**-**
hsa-miR-6767-5p	0.99	0.988	B
hsa-miR-6775-3p	1.00	0.995	B
hsa-miR-6824-5p	−1.01	0.975	-
hsa-miR-7641	1.00	0.997	B

A, the mean expression of these miRNAs is relatively low (<10) in one of the two groups and is reasonably higher in the other group (>10); B, the mean expression of these miRNAs is relatively low (<10) in both groups; in cases where there is “-” in the last column, the mean expression of the miRNA is relatively the same in both groups, and therefore, GeneGlobe has not provided any special comment. ^a^ Wald Test.

**Table 4 ijms-24-10884-t004:** Expression of urinary miRNA hsa-miR-875-5p, proposed as an endogenous normalizer, before and five days after the first cisplatin chemotherapy cycle (mean ± standard deviation), according to sequencing and qRT-PCR techniques.

**hsa-miR-875-5p**	**Expression with RNA-Seq technique** (mean ± standard deviation, reads)					
	**Participants with cisplatin-induced nephrotoxicity** **(*n* = 6)**		**Participants without cisplatin-induced nephrotoxicity** **(*n* = 5) ^a^**		**Fold change**	***p*-value ^b^**
**Baseline**	14.67 ± 10.39		38.83 ± 51.15		1.00	0.999
	**Expression with qRT-PCR technique** (mean ± standard deviation)					
	**Ct (*n* = 49)**			**Fold change** **(*n* = 49)**		
	**Participants with cisplatin-induced nephrotoxicity** **(*n* = 24)**	**Participants without cisplatin-induced nephrotoxicity** **(*n* = 25)**	***p*-value ^c^**			***p*-value ^c^**
**Baseline**	23.43 ± 1.26	23.63 ± 1.24	0.5354	1.01		0.9724
**D5**	23.08 ± 1.33	22.76 ± 1.42	0.4418	−1.43		0.9722

Ct, threshold cycle; D5, 5th day after the first cycle of cisplatin chemotherapy; n, absolute number of patients; ^a^ One of the participants belonging to the group without nephrotoxicity was excluded from these analyses because he had cel-miR-39 expression > or < 2 standard deviations in qRT-PCR (cel-miR-39 was used as an exogenous control). ^b^ Wald test. ^c^ Mann–Whitney test.

**Table 5 ijms-24-10884-t005:** Expression of urinary miRNA hsa-miR-4302, proposed as an endogenous normalizer, before and five days after the first cisplatin chemotherapy cycle (mean ± standard deviation), according to sequencing and qRT-PCR techniques.

**hsa-miR-4302**	**Expression with RNA-Seq technique** (mean ± standard deviation, reads)					
	**Participants with cisplatin-induced nephrotoxicity** **(*n* = 6)**		**Participants without cisplatin-induced nephrotoxicity** **(*n* = 5) ^a^**		**Fold change**	***p*-value ^b^**
**Baseline**	11.83 ± 7.41		23.17 ± 19.75		0.99	0.987
	**Expression with qRT-PCR technique** (mean ± standard deviation)					
	**Ct (*n* = 49)**			**Fold change** **(*n* = 49)**		
	**Participants with cisplatin-induced nephrotoxicity** **(*n* = 24)**	**Participants without cisplatin-induced nephrotoxicity** **(*n* = 25)**	***p*-value ^c^**			***p*-value ^c^**
**Baseline**	26.49 ± 1.51	25.34 ± 1.98	**<0.05**	−2.69		0.1135
**D5**	26.09 ± 1.42	25.85 ± 2.24	0.9644	−4.63		0.4936

Ct, threshold cycle; D5, 5th day after the first cycle of cisplatin chemotherapy; n, absolute number of patients; ^a^ One of the participants belonging to the group without nephrotoxicity was excluded from these analyses because he had cel-miR-39 expression > or < 2 standard deviations in qRT-PCR (cel-miR-39 was used as an exogenous control). ^b^ Wald test. ^c^ Mann–Whitney test.

## Data Availability

The datasets generated and/or analyzed during the current study are available from the Research Data Repository of the University of Campinas, https://redu.unicamp.br/dataset.xhtml?persistentId=doi:10.25824/redu/XT1BEY, accessed on 1 June 2023.

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
