# Peer review of "Data Normalization of Urine miRNA Profiling from Head and Neck Cancer Patients Treated with Cisplatin"

_ijms, 2023, doi:10.3390/ijms241310884_

Round 1

Reviewer 1 Report

I thank the authors for the interesting manuscript, entitled “Data normalization of urine miRNA profiling from head and 2 neck cancer patients treated with cisplatin” which aims at identifying one miRNA reliable as a normalizer in patients affected by head and neck cancer, even though it is not easy to find a miRNA suitable for this aim.

Following my comments and suggestion.

Results.

Paragraph 2.1: Lines79-83: which grade of nephrotoxicity? Might the authors please explain?

Figure 1: Did the authors check miRNA levels before starting the treatment with cisplatin? Just to have a sort of basal level.

Table 1: I am a bit hesitant about calculating the p-value for these parameters in only 6 patients: might the authors write something about that? Is there any reason?

Table 3: In the last column on the right (Comments by the Gen...): where is it not indicated A or B means that the miRNAs are expressed in all the samples? See criteria in lines 18-21.

Paragraph 2.3: Did the authors perform validation only for candidate miRNAs?

Table 4: Why the authors did not indicate the fold change for qRT-PCR?

Table 5: I am not so sure about the variation of miR-4302 as the state in lines 186-188, might the authors explain?

Figure 3: The SD are very high, did the authors think about working on a higher number of patients to increase the data and improve the SD?

Discussion.

Lines 35-36 and also Paragraph 2.2: the authors mention a high throughput assay to get an overview of miRNA expression, although it is not clear to me whether they used a panel for miRNA sequencing or something similar. Would please explain? Moreover, how many miRNAs did they assess in their cohort of patients?

Lines 61-63: The authors state that “the expression of hsa-miR-4302 varied significantly between the two studied 61 groups at the baseline period”, nevertheless there is no statistical significance between the groups (Figure 3). Might the authors please explain?

Materials and Methods.

Paragraph 4.2: Line 108: which stage of tumour? Lines 109-112: were these patients subjected

to surgery?

Conclusions.

It is not clear to me which miRNA between miR-875-5p and miR-4302 would be the best normalizer, assuming that there is one miRNA reliable for this aim.

As a general comment, I would suggest the authors increase the cohorts to improve SD.

In general, I would suggest a minor English revision to correct some typos. Following some indications.

Line 39: ... to select. Please revise.

Line 149: in bold would be better. Please revise.

Line 180: Table... show, is correct. Please revise.

Author Response

Dear Editor and Reviewers,

            We very much appreciated the considerations about our work, which helped us improve the manuscript. We reviewed the suggested changes in the text and answered the questions. You can find your comments addressed below. To make it easier for you to read, we leave all changes in the manuscript highlighted by using the "Track Changes" function in Microsoft Word, or  (when more appropriate) we leave a comment explaining the change. Please, do not hesitate to contact us if there are any issues left behind or new comments and suggestions.

Kind regards,

Nadine de Godoy Torso.

Reviewer 2 Report

De Godoy Torso et al. in their manuscript highlighted the difficulties of selecting a suitable endogenous gene for miRNA data normalization, especially when deriving from biofluids. They designed selection criteria of a stable candidate reference gene from smallRNAseq data and accurate normalization strategy;  they tried to validate the selected normalizers by RT-qPCR and reported 1 among the 4 selected miRNAs as sufficiently stable to be used as reference. Their work is of interest, because to date there is the lack of studies for the investigation of candidate normalizers for urinary miRNAs.

Author Response

Dear Editor and Reviewer,
            We very much appreciated your considerations about our work, and we would like to express our gratitude for your time and attention in helping us improve it. To make it easier for you to read, we leave all changes in the manuscript highlighted by using the "Track Changes" function in Microsoft Word. Please, do not hesitate to contact us if there are any issues left behind or new comments and suggestions.

Kind regards,

Nadine de Godoy Torso.

Round 2

Reviewer 1 Report

After this round of revision, I think the manuscript has been improved.

Nevertheless, I would suggest the authors not to put the p-values in Table 1, since there is no statistical difference between the groups. In their reply form, the authors want to highlight that the patients do not differ in their clinical characteristics, but only in their renal function, so I don't think such high and not significant p-values would be relevant for the reader.

Author Response

Thanks for the suggestion; we made the change in Table 1 and the revised manuscript was sent again.

Kind regards,

Nadine de Godoy Torso.